# Incorporating Expert Prior Knowledge for Oral Lesion Recognition

**Camille Besombes**[*,†]        **Adeetya Patel**[*,†]        **Sreenath Madathil**[†]

## Abstract

External information may improve predictive accuracy and uncertainty in medical image recognition. For example in oral lesion recognition, some lesion types are implausible to occur at certain anatomical locations. We propose a strategy to induce prior knowledge about such correlations using an additional loss term that optimizes for plausible lesion types given an anatomical location. Our results suggest an improvement in model calibration, a reduction in the predicted number of implausible classes, and improved uncertainty estimation for implausible predictions.

## 1 Introduction

Expert clinicians use prior knowledge on the distribution of lesion types to diagnose oral lesions in addition to the visual features of a case. One such crucial information is the anatomical location of the lesion. There are certain lesion types that rarely occur in certain anatomical locations. In limited data settings, the observed class distribution does not reflect the real-life distribution and thus the observed correlation structure between anatomical site and lesion type is biased. This leads deep learning models to predict implausible classes, or never predict some lesions for some anatomical site.

The current work focuses on methods to incorporate expert knowledge about the correlation structure between anatomical locations and lesion types into the model. Specifically, the extreme case where certain lesion types are implausible to occur at certain anatomical locations (See Fig. 2 in appendix). Furthermore, we investigate the implication of the prediction uncertainty using the approximate Bayesian method, Monte Carlo DropConnect (MC-DropConnect) (Mobiny et al., 2021).

**Contributions** of this work is a general method to incorporate expert knowledge during training. The proposed method has been shown to effectively reduce the number of implausible predictions and improve different metrics compared to the baseline method. Furthermore, our experiments demonstrate that the method yields better uncertainty estimates for implausible predictions.

## 2 Method

The MC-DropConnect setting approximates Bayesian inference by activating dropout and its variant DropConnect layers at training and inference time. As a result, a different subset $\theta_t$ of the network's weights is activated at each forward pass in the network and allows to sample the model posterior distribution for uncertainty quantification.

Given a dataset $\mathcal{D} = (\mathbf{x}, \mathbf{y})$ where $\mathbf{x}$ and $\mathbf{y}$ represent the input image and corresponding target vector, respectively. We define a neural network $\mathcal{M}$ that predicts output probabilities $\hat{\mathbf{y}} = \mathcal{M}_{\theta_t}(\mathbf{x})$. To account for anatomical constraints, we construct a set of implausible lesions $\omega$ given the anatomical site of the lesion present in the image based on the prior external knowledge shown in Fig. 2. To train the model, in addition to the cross-entropy loss ($CE$), we add a penalization term that computes mean squared error ($MSE$) between the output probabilities $\hat{\mathbf{y}}$ and one-hot encoded target-vector $\mathbf{y}$

---

[*]These authors contributed equally to this work.

[†]Faculty of Dental Medicine and Oral Health Sciences, McGill University, Montreal, Canada.

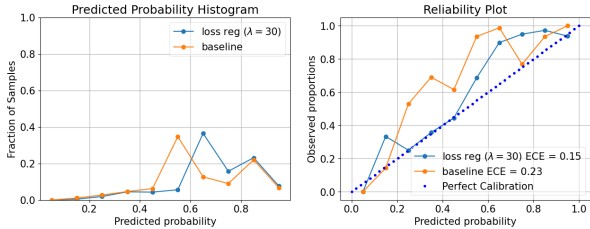

Figure 1: Reliability plot comparing our method and the baseline. The left panel is the confidence histogram, the right panel is the calibration curve along with Expected Calibration Error (ECE). We observe that ECE is substantially reduced for our method.

only for implausible lesion types given in $\omega$. The penalization term is defined as:

$$L_{MSE} = \frac{1}{N} \sum_{i=1}^{N} \sum_{c=1}^{C} (y_{i,c} - \hat{y}_{i,c})^2 * \mathbb{1}_{(c \in \omega_i)},$$

where $C$ is the number of classes and $N$ is the number of samples. We note that the set of implausible lesion $\omega$ changes per sample based on the anatomical location of the lesion. Finally, the loss function of our model can be written $L = L_{CE} + \lambda L_{MSE}$ where $\lambda$ is the weight factor.

## 3 EXPERIMENTS AND RESULTS

**Dataset**   The dataset includes 2700 images featuring 16 distinct oral lesion types, each tagged with an anatomical location label. A matrix of plausible anatomical locations for each class was created by consulting with oral pathologists and literature shown in Fig. 2. It can be seen that most lesion types can occur at multiple anatomical sites, and three of them are linked to a unique anatomical site. Unfortunately, we have chosen not to release the dataset publicly due to privacy concerns.

**Discussion of results**   Table1 shows a comparison between a baseline model and four versions of the proposed method with different loss weights $\lambda \in (5, 10, 30, 50)$. One hundred stochastic forward passes were considered to sample the model posterior and compute different metrics for each experiment. Implausible predictions represent the number of predictions of a lesion type incoherent with the anatomical site visible in the image. The number of implausible predictions is mainly decreased, and calibration is improved for high weights associated with the penalization term. Mutual information can be interpreted as a measure of model uncertainty, one should note the higher mutual information values for implausible predictions by our method compared to the baseline. It shows an improved quality of uncertainty estimation induced by the penalization term. Finally, Fig. 1 shows a significant improvement in the model calibration averaged over all the classes.

Table 1: Comparison of baseline and our methods with different values of $\lambda$. MI stands for Mutual Information, SBS for Scaled Brier Score and ECE stands for Expected Calibration Error. The details about different metrics can be found in Appendix A.7.

| Methods | Implausible preds ($\downarrow$) | MI for implausible preds ($\uparrow$) | SBS ($\uparrow$) | ECE ($\downarrow$) |
|---|---|---|---|---|
| Baseline ($\lambda = 0$) | 13/622 | 0.10 | 0.690 | 0.23 |
| Ours ($\lambda = 5$) | 11/622 | **0.16** | 0.696 | 0.19 |
| Ours ($\lambda = 10$) | **8/622** | 0.15 | 0.715 | 0.20 |
| Ours ($\lambda = 30$) | 11/622 | 0.13 | **0.718** | **0.15** |
| Ours ($\lambda = 50$) | **8/622** | 0.15 | 0.711 | 0.18 |

## 4 CONCLUSION

In this work, we presented a method to incorporate expert knowledge into deep learning models through a penalization term. We demonstrate that it can improve model performance by decreasing the number of predictions incoherent with external domain knowledge. Additionally, our method seems to have a beneficial effect on the training by improving uncertainty estimation and calibration for oral lesion recognition tasks. This approach can be extended to other tasks where similar correlation structures are observed.

URM STATEMENT

The authors acknowledge that at least one key author of this work meets the URM criteria of ICLR 2023 Tiny Papers Track.

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

## A  APPENDIX

### A.1  THE RELATIONSHIP BETWEEN MC-DROPCONNECT AND OUR METHOD

It is important to note that using MC-DropConnect is not necessary for our method. The main purpose of employing the approximated Bayesian method is to quantify uncertainty in lesion classification and be able to see the effect of our proposed method on uncertainty quantification. To serve this purpose, other uncertainty quantification methods can also be used with our method. The reduction of implausible predictions is directly associated with high certainty within plausible classes given an anatomical site of the lesion.

### A.2  CHOICE OF METRICS AND POTENTIAL ALTERNATIVES

The metrics chosen in our paper aim to show an improvement in uncertainty quantification in the model, rather than one in accuracy. This is because reducing the number of implausible predictions may not necessarily result in increased accuracy, since some wrong predictions are still possible among the plausible classes given an anatomical site. Due to space constraints, we only report the most relevant metrics, but it is also important to monitor accuracy and other classification metrics to fully assess the method and the effect of the $\lambda$ hyperparameter on the model's performance.

The Expected Calibration Error was picked as a relevant metric for its strong connection with uncertainty quantification and interpretability of the model. A lower ECE indicates higher confidence in the model's predictions. In a study by Maier-Hein et al. (2022), the medical community expressed the need for additional calibration metrics to be included in scientific literature. Moreover, the Scaled Brier Score (SBS) is presented as a means to evaluate the balance between the discrimination power and calibration of a classifier. Discrimination power refers to the classifier's ability to accurately distinguish between different lesions, while calibration measures how well its predicted probabilities align with the observed probabilities. The SBS provides insights into this balance, allowing for a comprehensive evaluation of the classifier's performance in terms of accurate classification and well-calibrated probability estimates.

### A.3  EFFECTS OF MODIFYING THE $\lambda$ PARAMETER

The sensitivity analysis conducted on the $\lambda$ parameter aims to determine the ideal balance between the cross-entropy and mean squared error (MSE) losses. For instance, if $\lambda$ values are too large, the

neural network may primarily focus on avoiding implausible predictions without adequately learning lesion classification.

In Table. 1, it is evident that the number of implausible predictions does not follow a linear relationship with the $\lambda$ parameter. It's important to keep in mind that the count of implausible predictions only considers the highest probability among all classes, without taking into account the reduced softmax scores for implausible classes. On the other hand, SBS is directly calculated using the softmax scores and is optimal when $\lambda$ is set to 30. This finding aligns with the imposed constraint and strikes a good balance between discrimination and calibration for the classifier.

### A.4    DATA COLLECTION AND SPECIFICS

The dataset was collected by oral pathologists as part of the routine clinical examination of patients with oral lesions and is a part of their electronic health record. The images were captured using a digital camera of varying quality, under dental clinic lighting. The ground truth labels are obtained from pathology reports following the gold-standard diagnosis technique of biopsy or expert clinician's diagnosis. Due to the limitation of access to the oral cavity, images of specific anatomical locations (e.g. palate) can only be taken in a limited number of ways. Further, images may contain identifiable information about the patient (e.g., parts of eyes, skin color, dentition) and hence we, unfortunately, are not able to provide an open dataset. Our protocol was approved by Institutional Review Board.

**Data leakage**    One of the key challenges in oral lesion classification is the issue of data leakage, which occurs when the algorithm learns by using information from the data that should not be used for the task at hand such as the shape of teeth etc. This can lead to overfitting and poor generalization of the model. To avoid data leakage, we created clusters of images such that the images of the same patient are kept in the same cluster. We make use of EXIF data from the images and other information such as patient names, lesion types, and anatomical sites to make clusters of images. We assign these clusters to either training or test sets to avoid data leakage.

### A.5    CONSTRUCTION OF EXPERT KNOWLEDGE MATRIX OF IMPLAUSIBLE LESIONS ($\omega_i$)

An initial matrix was constructed after a comprehensive review of the literature on the reported occurrence of each type of oral lesion. The literature included peer-reviewed publications, pedagogical textbooks, and other reports. This initial matrix was reviewed for correctness and completeness by oral pathology experts. The final matrix is presented in Fig. 2. For each training and test sample, the column of this matrix corresponding to the anatomical location of the lesion in the image is chosen as the $\omega_i$.

### A.6    TRAINING DETAILS

For all our experiments, we use an ImageNet Deng et al. (2009) pretrained efficientnet-b5 Tan & Le (2019). We finetune the full network weights for 400 epochs using RMSprop optimizer with an initial learning rate of $10^{-3}$. We use a dropout and dropconnect rate of 0.5.

### A.7    METRICS

**Brier score** (Brier et al., 1950) is equivalent to an MSE between the predicted vector and the one-hot encoded ground truth. The Scaled Brier Score (SBS) is a skill score that allows comparison with a reference Brier score ($BS_{ref}$) such as $SBS = 1 - \frac{BS}{BS_{ref}}$. The reference was chosen to be a non-informative model that outputs the marginal proportion of classes for every sample. The scaled Brier score has the added advantage that it is not affected by the marginal distribution of the true class.

**Expected Calibration Error**: Evaluating calibration consists of measuring the statistical consistency between the predictive distribution and the observed proportion *i.e.* the accuracy of the predictive distribution. Expected Calibration Error is a binning method where the predictive probability $y_{pred_i}$ is computed and grouped into $M$ bins $b_1, ..., b_M$. Then, calibration of the single bins is eval-

uated by setting the average bin predictive probability:

$$pp(b_m) = \frac{1}{|b_m|} \sum_{s \in b_m} y_{pred_s}$$

in relation with the observed proportions:

$$op(b_m) = \frac{1}{|b_m|} \sum_{s \in b_m} \mathbb{1}(\arg\max(y_{pred_s}) = \arg\max(y_s))$$

The model is calibrated when $pp(b_m) = op(b_m)$, the Expected Calibration Error gives a quantitative evaluation defined by:

$$ECE = \sum_{m=1}^{M} \frac{|b_m|}{N} |pp(b_m) - op(bm)|$$

**Mutual Information (MI)** is defined by the amount of information gained about the model parameters by receiving a test sample $\mathbf{x}$ and its corresponding true label $\mathbf{y}$.

$$I(\mathbf{y}, \theta | \mathbf{x}, \mathcal{D}) = H(\mathbf{y}|\mathbf{x}, \mathcal{D}) - \mathbb{E}_{p(\theta|\mathcal{D})} H(p(\mathbf{y}|\mathbf{x}, \theta))$$

where $H$ is the predictive entropy that can be approximated by Bayesian approximation using Monte Carlo sampling

$$H(\mathbf{y}|\mathbf{x}, \mathcal{D}) = - \sum_c p(\mathbf{y} = c|\mathbf{x}, \mathcal{D}) \log p(\mathbf{y} = c|\mathbf{x}, \mathcal{D}) \approx - \sum_c p_{MC}(\mathbf{y} = c|\mathbf{x}) \log p_{MC}(\mathbf{y} = c|\mathbf{x})$$

where $p_{MC}$ is the average of the predictive probability over T Monte Carlo forward passes. It yields

$$I(\mathbf{y}, \theta | \mathbf{x}, \mathcal{D}) \approx H(\mathbf{y}|\mathbf{x}, \mathcal{D}) - \sum_c \frac{1}{T} \sum_{t=1}^{T} p(\mathbf{y} = c|\mathbf{x}, \theta_t) \log p(\mathbf{y} = c|\mathbf{x}, \theta_t)$$

## A.8    ADDITIONAL RESULTS

We show reliability plots for each anatomical site in  Fig. 3 to 10. We observe that the expected calibration error (ECE) is decreased for almost all anatomical sites except retromolar trigone as shown in  Fig. 9. It is known that ECE is sensible to the number of samples considered, which is low for retromolar trigone location.

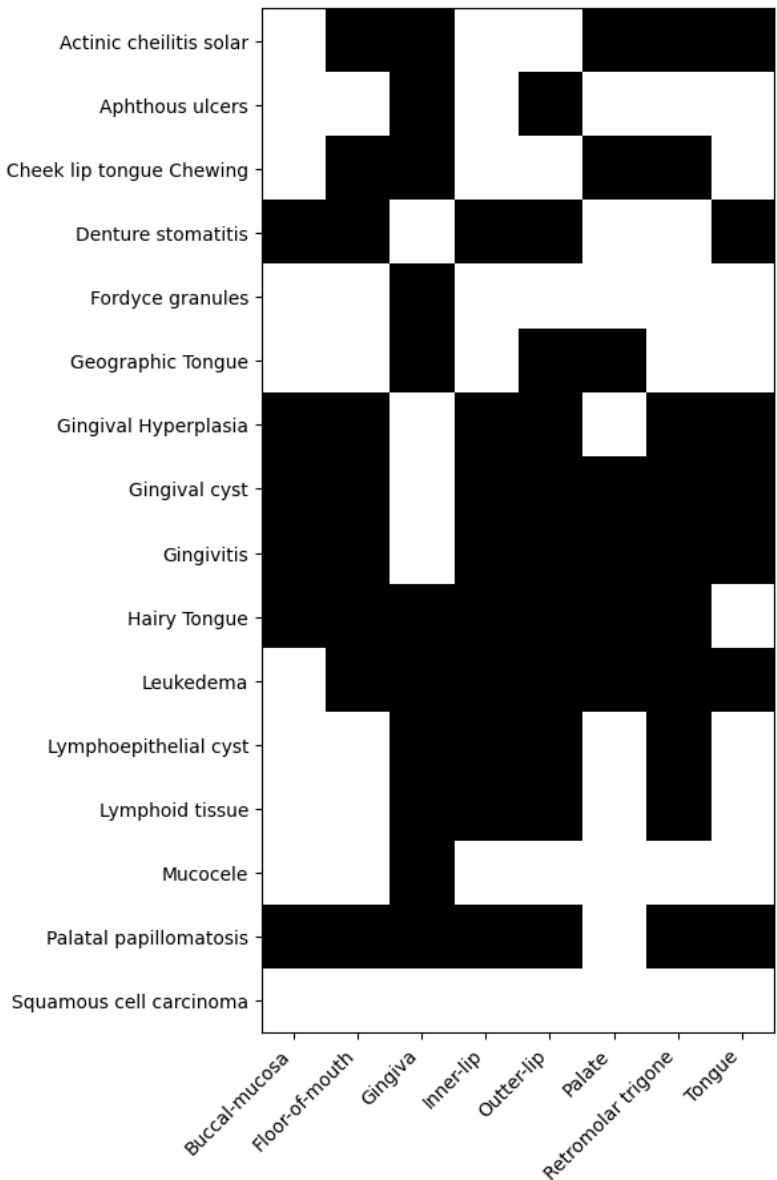

Figure 2: Anatomical site and lesion occurrence plausibility. The black square represents the implausibility of the lesion type to occur on the anatomical site. The white square represents plausibility.

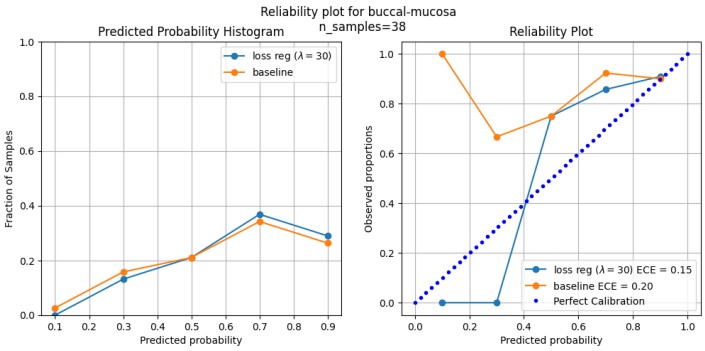

Figure 3: Reliability plot for the buccal-mucosa anatomical site.

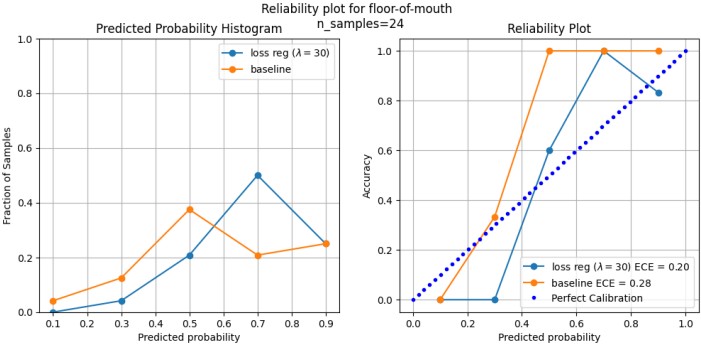

Figure 4: Reliability plot for the floor of mouth anatomical site.

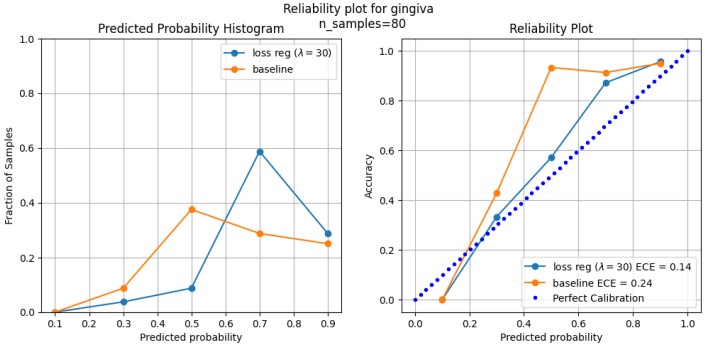

Figure 5: Reliability plot for gingiva anatomical site.

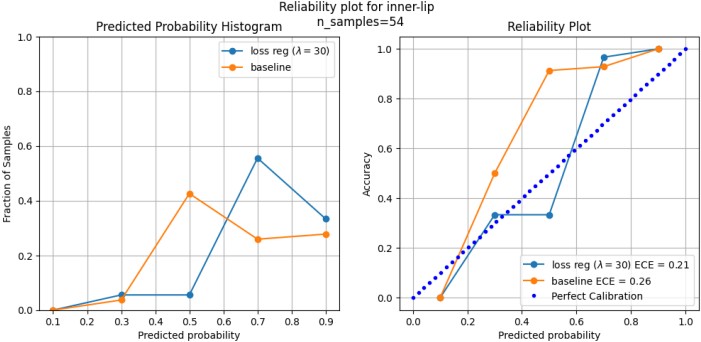

Figure 6: Reliability plot for the inner-lip anatomical site.

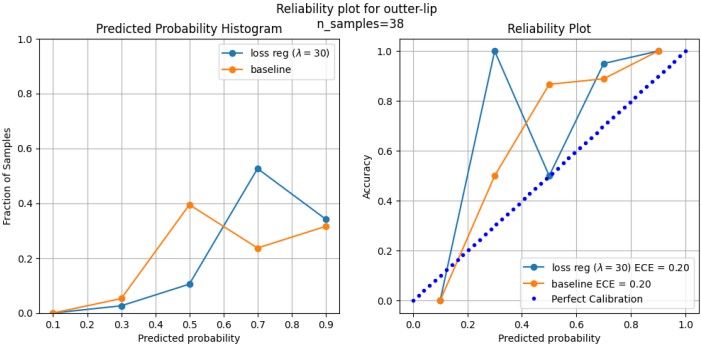

Figure 7: Reliability plot for the outer-lip anatomical site.

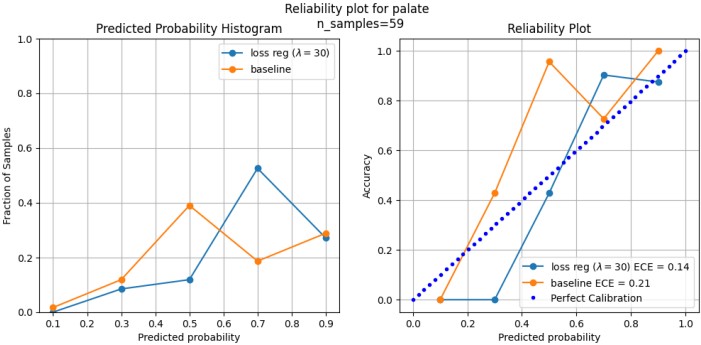

Figure 8: Reliability plot for palate anatomical site.

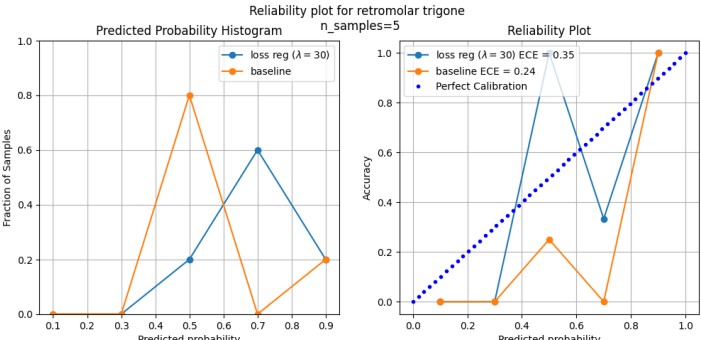

Figure 9: Reliability plot for retromolar trigone anatomical site.

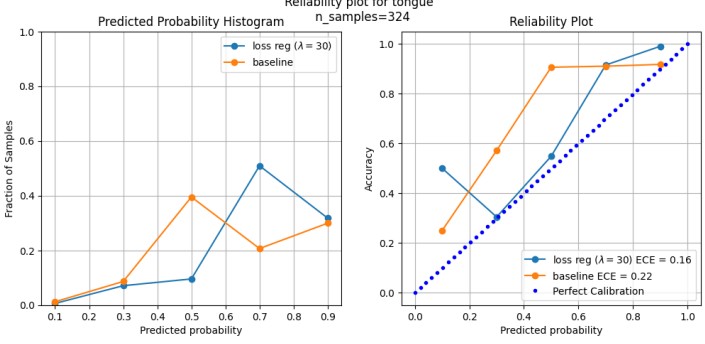

Figure 10: Reliability plot for tongue anatomical site.

