# OpenReview forum: "Incorporating Expert Prior Knowledge for Oral Lesion Recognition"
_ICLR.cc/2023/TinyPapers — Submitted to Tiny Papers @ ICLR 2023_

### Official Review · Reviewer_R95M · 2023-03-27

**Confidence:** 3

**Summary Of Contributions:**

The paper presents a novel training scheme to improve the performance of oral lesion recognition by imposing expert prior knowledge to a total loss while training neural network models. The proposed prior is a mean squared error (MSE) loss between predictions and ground truth of lesion types. Since the proposed regularized loss constraints to the lesion types and locations, the proposed scheme achieves better scaled brier scores and lower implausible predictions.

**Rating:**

High Potential (HP): a submission which meets the reviewing criteria and has potential to make an impact on the field

**Strengths And Weaknesses:**

Strengths:
1. The paper was compressive and well-written. It contains unambiguous writing via separate sections having the introduction, approach, and experimentation. It is easy for readers to understand the problem that authors try to solve and what their contribution is.
2. The authors propose a simple but effective regularization loss that imposes expert prior knowledge to constrain the total loss for model training. Since the regularization loss penalizes the difference between predictions and ground truth of implausible lesion types, the proposed scheme achieves better Mutual Information and lower Implausible predictions.

Weaknesses:
1. The connection between the proposed regularization loss and Monte Carlo DropConnect (MC-DropConnect) is vague. The MC-DropConnect technique has been used to enhance model calibration in deep neural networks. The paper only mentions their proposed MSE loss but does not address the connection between the proposed loss and MC-DropConnect. Do the authors also consider MC-DropConnect while proposing the MSE loss? Or do the authors propose the MSE loss inspired by MC-DropConnect?
2. The evaluation metrics are not clear. The paper reports multiple measurement metrics for experiment comparisons. Since the proposed loss does not have a strong connection with MC-DropConnect, what do the reasons include Excepted Calibration Error in their experiment?

**Suggested Changes:**

Suggested Changes:
1. The paper can have descriptions for the connection between MC-Dropconnect and the proposed MSE loss to increase paper readability in the method section.
2. The authors can also also consider other evaluation metrics or explain why they choose the existing metrics presented in the paper.

---

### Official Review · Reviewer_9vpk · 2023-03-29

**Confidence:** 4

**Summary Of Contributions:**

In this paper, the authors focus on the problem of oral lesion recognition. To this end, this paper proposes a general method to incorporate expert knowledge during training to reduce the number of implausible predictions based on anatomical location.

**Rating:**

Clear, Correct, and Reproducible (CCR): a submission which meets the reviewing criteria

**Strengths And Weaknesses:**

**Strength**
* S1: The paper presents results which show that the method improved performance as compared to the baseline
* S2: The appendix clearly explain each metric and presents additional results to support the claims made in the paper
* S3: The paper clearly shows the hyperparameters used to train the model.
* S4: The data collection methods have been explained clearly

**Weaknesses**
* W1: The paper tries different values of the parameter λ. However, for some values, the number of implausible preds is the same (for example λ = 10 and λ = 50). The discussion of results does not comment on this behavior.
* W2: The reason for using MCDropConnect has not been clearly explained. Moreover, the paper does not address why the loss function is suitable for use with MCDropConnect


**Suggested Changes:**

* SC1: Brief description should be added in the *Discussion of Results* section to explain the effect of changing λ on the metrics.

---

### Author Response · Authors · 2023-05-31
**Opt-in statement**

The authors hereby express their voluntary decision to opt-in and participate in the archiving process.

---

### Meta-Review · Area_Chair_cg2T · 2023-04-07

**Recommendation:** Invite to present
**Confidence:** 4

**Metareview:**

The paper proposes a method for incorporating expert knowledge during training to reduce the number of implausible predictions based on anatomical location. The method imposes a mean squared error loss between predictions and ground truth of lesion types as a regularization loss, which constraints the lesion type and location. The proposed scheme achieves better-scaled brier scores and lower implausible predictions compared to the baseline. The paper is well-written and presents a comprehensive approach to the problem. However, the reviewers suggest improvements in the connection between the proposed regularization loss and Monte Carlo DropConnect, and the evaluation metrics.

**Summary:**

The paper proposes a method for incorporating expert knowledge during training to improve the performance of oral lesion recognition based on anatomical location. The method imposes a mean squared error loss between predictions and ground truth of lesion types as a regularization loss, which constraints the total loss for model training. The proposed scheme achieves better scaled brier scores and lower implausible predictions compared to the baseline.

**Comments And Feedback To The Authors:**

The authors should address the reviewers' suggestions to enhance the clarity and comprehensibility of the paper. Specifically, the authors should add descriptions for the connection between MC-Dropconnect and the proposed MSE loss in the method section to increase paper readability. Additionally, the authors should consider other evaluation metrics or explain why they choose the existing metrics presented in the paper. Lastly, the authors should add a brief description in the discussion section to explain the effect of changing λ on the metrics.

**Reason For Not Giving A Higher Recommendation:**

We suggest improvements in the connection between the proposed regularization loss and Monte Carlo DropConnect, and the evaluation metrics. Some somethings are not explained, eg. exact scores for lambda = 10 and 50 in table 1.

**Reason For Not Giving A Lower Recommendation:**

The paper presents a novel method that achieves improved performance compared to the baseline. The paper is well-written, and the appendix clearly explains each metric, while the data collection methods have been explained clearly. It is easy to understand the simple but effective contirubtions

---

> ### Author Response · Authors · 2023-05-31
> **Answer to reviewers**
>
> First of all, we would like to thank all the reviewers for underlining the clarity of our paper and their constructive feedback that helped us improve our work. You will find a detailed answer for each comment below:
>
>
> ## The connection between MC-DropConnect and our method
> ---
> Reviewer R95M:
>
> > “1. The connection between the proposed regularization loss and Monte Carlo DropConnect (MC-DropConnect) is vague. The MC-DropConnect technique has been used to enhance model calibration in deep neural networks. The paper only mentions their proposed MSE loss but does not address the connection between the proposed loss and MC-DropConnect. Do the authors also consider MC-DropConnect while proposing the MSE loss? Or do the authors propose the MSE loss inspired by MC-DropConnect?”
>
> Reviewer 9vpk:
>
> > “• W2: The reason for using MCDropConnect has not been clearly explained. Moreover, the paper does not address why the loss function is suitable for use with MCDropConnect”
>
> Thank you for letting us know about the lack of clarity between our proposed method and the use of the MC-DropConnect framework in our work.
>
> It is important to note that using MC-DropConnect is not necessary for our method. The main purpose of employing the approximated Bayesian method is to quantify uncertainty in lesion classification and be able to see the effect of our proposed method on uncertainty quantification. To serve this purpose, other uncertainty quantification methods can also be used with our method. The reduction of implausible predictions is directly associated with high certainty within plausible classes given an anatomical site of the lesion.
>
> This precision was added to Appendix A.1.
>
>
> ## Justification for choice of metrics and potential use of others
> ---
> Reviewer R95M:
>
> > “1. The evaluation metrics are not clear. The paper reports multiple measurement metrics for experiment comparisons. Since the proposed loss does not have a strong connection with MC-DropConnect, what do the reasons include Excepted Calibration Error in their experiment?”
>
> The metrics chosen in our paper aim to show an improvement in uncertainty quantification in the model, rather than one in accuracy. This is because reducing the number of implausible predictions may not necessarily result in increased accuracy, since some wrong predictions are still possible among the plausible classes given an anatomical site. Due to space constraints, we only report the most relevant metrics, but it is also important to monitor accuracy and other classification metrics to fully assess the method and the effect of the $\lambda$ hyperparameter on the model's performance.
>
> The Expected Calibration Error was picked as a relevant metric for its strong connection with uncertainty quantification and interpretability of the model. A lower ECE indicates higher confidence in the model's predictions. In a study by \citet{Maier-Hein2022}, the medical community expressed the need for additional calibration metrics to be included in scientific literature. Moreover, the Scaled Brier Score (SBS) is presented as a means to evaluate the balance between the discrimination power and the calibration of a classifier. Discrimination power refers to the classifier's ability to accurately distinguish between different lesions, while calibration measures how well its predicted probabilities align with the observed probabilities. The SBS provides insights into this balance, allowing for a comprehensive evaluation of the classifier's performance in terms of accurate classification and well-calibrated probability estimates.
>
> This precision was added in Appendix A.2.
>
>
> ## Effect of changing the lambda parameter
> ---
> Reviewer 9vpk:
>
> > “• W1: The paper tries different values of the parameter λ. However, for some values, the number of implausible preds is the same (for example λ = 10 and λ = 50). The discussion of results does not comment on this behavior.”
>
> The sensitivity analysis conducted on the $\lambda$ parameter aims to determine the ideal balance between the cross-entropy and mean squared error (MSE) losses. For instance, if $\lambda$ values are too large, the neural network may primarily focus on avoiding implausible predictions without adequately learning lesion classification.
>
> In Table.1, it is evident that the number of implausible predictions does not follow a linear relationship with the $\lambda$ parameter. It's important to keep in mind that the count of implausible predictions only considers the highest probability among all classes, without taking into account the reduced softmax scores for implausible classes. On the other hand, SBS is directly calculated using the softmax scores and is optimal when $\lambda$ is set to 30. This finding aligns with the imposed constraint and strikes a good balance between discrimination and calibration for the classifier.
>
> Precision added in Appendix A.3.

---

### Decision · Program_Chairs · 2023-04-08

Invite to present